# Exploration of predictive factors based on oral and intestinal bacterial flora for treating patients with urothelial carcinoma

Yuichi Matsumoto[1]*, Yukihiro Hitaka [1], Hiroshi Hirata[1], Yoshiaki Yamamoto[2],
Keita Kobayashi [1], Naohito Isoyama[1], Toshio Matsubara[3], Kenji Watanabe [4],
Yoichi Mizukami[4], Shin Nakagawa[3], Katsuaki Mishima[5], Koji Harada[6], Koji Shiraishi[1]

**1** Department of Urology, Graduate School of Medicine, Yamaguchi University, Ube, Yamaguchi, Japan,
**2** Department of Urology, Sanyo-Onoda Municipal Hospital, Sanyo-Onoda, Yamaguchi, Japan, **3** Division
of Neuropsychiatry, Department of Neuroscience, Yamaguchi University Graduate School of Medicine,
1-1-1 Minami-Kogushii, Ube, Yamaguchi, Japan, **4** Center for Gene Research, Science Research Center,
Yamaguchi University, Ube, Yamaguchi, Japan, **5** Department of Oral and Maxillofacial Surgery, Graduate
School of Medicine, Yamaguchi University, Ube, Yamaguchi, Japan, **6** Department of Nursing, Faculty of
Health Sciences, Hiroshima Cosmopolitan University, 5-3-18 Ujina-nishi, Minami-ku, Hiroshima, Japan

* h.yukki@yamaguchi-u.ac.jp

journal.pone.0324814

Andhra Pradesh, INDIA

**Peer Review History:** PLOS recognizes the
benefits of transparency in the peer review
process; therefore, we enable the publication
of all of the content of peer review and
author responses alongside final, published
articles. The editorial history of this article is
available here: https://doi.org/10.1371/journal.
pone.0324814

## Abstract

### Purpose

The role of intestinal flora in carcinogenesis and chemotherapy efficacy has been
increasingly studied; however, comparisons between oral and intestinal flora remain
limited. Given the oral microbiome's role in systemic inflammation and immune modu-
lation, it may significantly influence cancer progression and response to immunother-
apy. This study aimed to identify the microbial changes in urothelial carcinoma (UC)
by analyzing oral saliva and stool samples from healthy individuals and UC patients.
We also examined the association between microbial composition and immune
checkpoint inhibitor (ICI) response.

### Methods

A total of 20 healthy individuals and 38 patients with UC were analyzed. Among
them, 27 patients with UC underwent ICI treatment. Oral saliva and stool samples
were analyzed for 16S rRNA sequences to assess bacterial composition. Operational
taxonomic units were generated, and phylogenetic analysis was performed using the
Illumina BaseSpace.

### Results

Patients with UC showed higher *Veillonellaceae* and *Prevotellaceae* levels in saliva
and stool, with lower levels of these bacteria associated with more prolonged overall

**Data availability statement:** The datasets generated and analyzed during the current study are available in the repository, persistent https://www.ncbi.nlm.nih.gov/geo/query/acc.cgi?acc=GSE293354. All original code is available at NCBI GEO under accession number GSE293354.

**Funding:** The author(s) received no specific funding for this work.

**Competing interests:** The authors have declared that no competing interests exist.

survival and progression-free survival, particularly *Veillonellaceae* in stool. A higher neutrophil-to-lymphocyte ratio correlated with increased levels of these bacteria.

## Conclusion

*Veillonellaceae* and *Prevotellaceae* are potential microbial biomarkers of survival outcomes and ICI efficacy in patients with UC. Non-invasive oral microbial sampling may facilitate personalized cancer treatment strategies.

---

## Introduction

Recent advances in microbiome research have revealed that the oral and intestinal microbiota are key regulators of various physiological processes, including immune regulation, inflammation, and even cancer progression [1–6]. Particularly, the intestinal microbiota has received considerable attention for its role in carcinogenesis and host response to cancer therapies, such as chemotherapy and immunotherapy [3,7]. The human microbiome comprises trillions of microorganisms that are crucial in maintaining homeostasis [2]. However, when this balance is disrupted, a condition known as dysbiosis occurs, which has been linked to the development of diseases, including urothelial carcinoma (UC) and renal cell carcinoma [8,9]. Although much research has focused on intestinal microbiota, the role of oral microbiota is comparatively underexplored despite its proximity to key digestive and respiratory pathways [3]. Recent studies have suggested that oral flora, such as intestinal flora, may affect cancer progression and treatment outcomes through immune modulation and metabolic interactions [5,6,10].

Both oral and gut microbiota are closely associated with systemic inflammation [11,12]. Specific bacterial populations trigger inflammatory responses, contributing to elevated neutrophil-to-lymphocyte ratio (NLR) [13]. The NLR, an inflammatory marker calculated as the ratio of neutrophils to lymphocytes, is widely recognized as a key prognostic factor in cancer [14]. Studies have shown that a high NLR is associated with cancer progression and a reduced treatment response because of its role in promoting inflammatory cytokine secretion and fostering an immunosuppressive tumor microenvironment [14]. Notably, elevated NLR levels have been correlated with reduced efficacy of immune checkpoint inhibitors (ICI) in cancer patients as well as poorer outcomes in terms of overall survival (OS) and progression-free survival (PFS) [15,16].

ICIs, the cornerstone of UC treatment, are significantly influenced by the gut microbiota composition [17–19]. Specific groups of oral and gut bacteria can enhance or diminish the efficacy of ICI treatment by influencing systemic inflammation and immune responses via NLR. For example, a high NLR reflects an inflammatory response; however, adjusting the balance of specific bacterial populations may reduce the NLR and improve treatment outcomes.

The aim of this study is to identify changes in the bacterial flora in urothelial carcinoma by comparing the oral and intestinal bacterial flora of healthy people and

patients with urothelial carcinoma (UC), and to study the relationship between the composition of the bacterial flora and the response to immune checkpoint inhibitors (ICI). Using 16S rRNA gene sequencing, we aim to characterize microbial shifts associated with cancer development and response to immunotherapy [20]. By identifying key bacterial patterns without bias toward specific taxa, this study seeks to contribute to the understanding of the microbiome's role in urothelial carcinoma and its potential implications for treatment strategies [21,22].

Furthermore, a deeper understanding of the interactions between these bacteria and the immune response will contribute to the development of biomarkers utilizing oral and gut bacterial flora and provide a pathway toward ICI therapy personalization [21,22]. We hope this study will lay the foundation for further improvements in cancer treatment and the ability to make optimal treatment choices for each patient.

## Materials and methods

### Study design and clinical samples

Samples were collected from 38 patients with metastatic UC (38 patients) treated with ICIs at Yamaguchi University Hospital between April 2020 and March 2022, before using ICIs. Table 1 shows the clinicopathological characteristics of the patients.

**Table 1. Patient's background.**

| | UC | NORMAL |
|---|---|---|
| | N = 38 | N = 20 |
| | n(%),n(range) | n(%),n(range) |
| **Factor/category** | | |
| **Gender** | | |
| Men | 28(73.7) | 4(20) |
| Women | 10(26.3) | 16(80) |
| **Age** | 74(60-89) | 61(46-82) |
| **Previous primary leision removal** | | |
| Yes | 21(55.3) | |
| No | 17(44.7) | |
| **Prior systemaic therapy before IO** | | |
| | 0 | 1(2.6) |
| 1 | 20(52.6) | |
| 2 | 6(15.8) | |
| 3 | 3(7.9) | |
| >4 | 0(0) | |
| no IO | 8(21.1) | |
| **metastatics sites** | | |
| 0 | 1(2.6) | |
| 1 | 26(68.4) | |
| 2 | 4(10.5) | |
| >3 | 7(18.4) | |
| **ECOG PS** | | |
| 0 | 13(34.2) | |
| 1 | 17(44.7) | |
| >2 | 8(21.1) | |

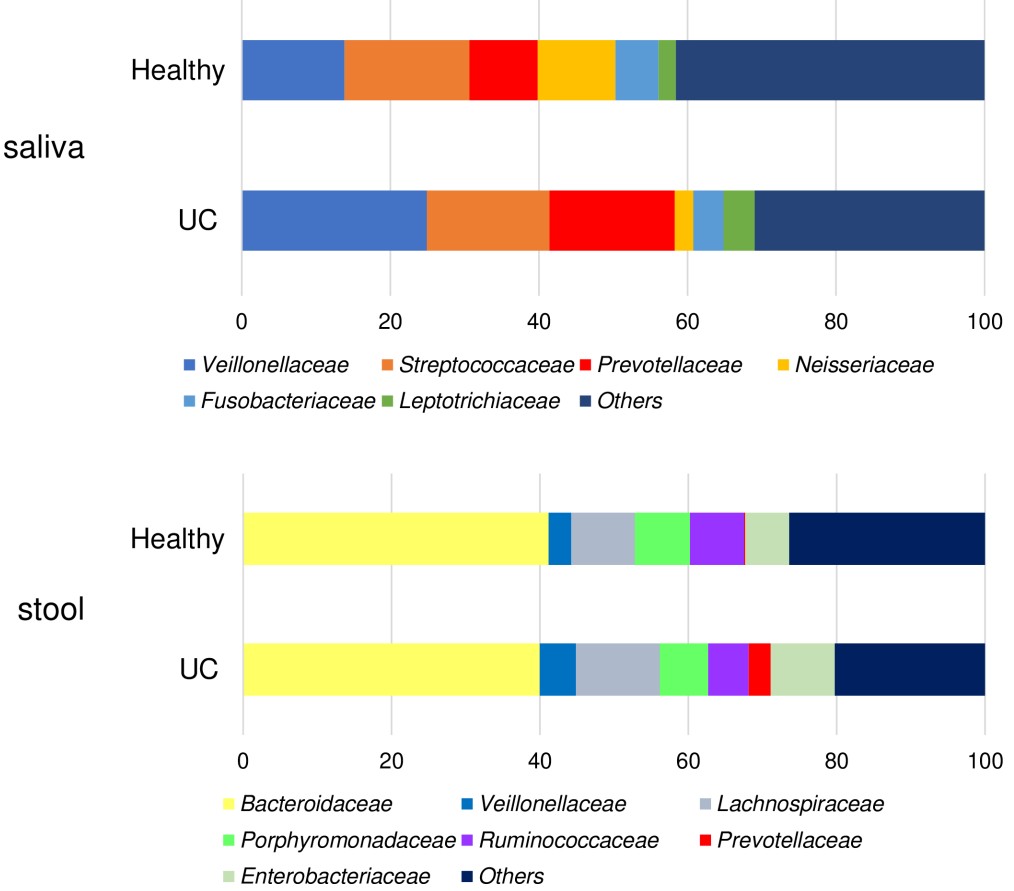

**Fig 1. Visualization of microflora sequencing results in urothelial carcinoma (UC) and healthy groups using a 100% horizontal stacked bar chart.** (a) Salivary bacterial genera composition. (b) Fecal bacterial genera composition.

During the same period, samples were collected from 20 healthy non carcinoma donors as controls. The oral and intestinal microflora of each patient were examined, and the distributions of oral and fecal microflora in healthy subjects and patients with cancer were retrospectively assessed. The proportions of the different stages of the obtained bacterial species were calculated and used in the analysis (Fig 1).

The bar chart shows the relative abundances of *Veillonellaceae*, *Streptococcaceae*, *Prevotellaceae*, and other bacterial families in saliva and stool samples from the UC and control groups. Each color represents a specific bacterial family. The X-axis indicates the sample group and the Y-axis indicates the percentage of abundance.

All procedures in studies involving human subjects were conducted in accordance with the ethical standards of the institutional and/or national research committee and with the Helsinki Declaration of 1964 and its subsequent amendments or equivalent ethical standards. The study received the approval of the Institutional Review Board (IRB) of Yamaguchi University Hospital (approval number H 27-023-3). No financial incentives were offered or offered for participation. Participants were recruited and samples were collected between 1 April 2020 and 31 March 2022. Written informed consent, informed consent, including agreement to the ethical provisions and online information, was obtained from all participants. For retrospective analysis of medical records and stored samples, data were fully anonymised before access. We accessed data from April 1, 2020 to October 31, 2024.

## DNA extraction from oral specimens

Oral bacteria were collected from the periodontal pockets and tongue tissue during oral care. In the edentulous jaw, we collected bacteria only from the tongue, whereas in the case of the teeth, we collected bacteria from the periodontal pockets. The QIAamp DNA Mini Kit (QIAGEN) was used to extract genomic DNA from the oral samples.

## DNA extraction from fecal samples

Fecal samples were collected by the participants themselves following the manufacturer's instructions. DNA was extracted from 200 mg of stool using a QIAamp DNA Stool Mini Kit (Qiagen, Hilden, Germany).

## 16S ribosomal RNA sequencing for gut and salivary microbiota genomics

After determining the concentration of genomic DNA extracted from the samples using Nanodrop one (Thermo Fisher Scientific), 6.25 ng DNA was added to 25 µl of 2x KAPA HS HiFi ready mix (KAPA Biosystem). The DNA mixture was amplified using a PCR thermal cycler (C1000 Touch Thermal Cycler, Bio-Rad) programmed with an initial denaturation of 3 minutes at 95°C, followed by 25 cycles of denaturation at 95°C for 30 seconds, annealing at the temperature of 72°C for 30 seconds, and extension at 72°C for 30 seconds with the following primer pair for the V3-V4 region of 16S rRNA. Forward primer: TCGTCGGCAGCGTCAGATGTGTATAAGAGACAGCCTACGGGNGGCWGCAG, 16S amplicon Reverse primer: TCTCGTGGGCTCGGAGATGTGTATAAGAGACAGGACTACHVGGGTATCTAATCCA.

The barcode and adaptor sequence were inserted into the products by the Nextera XT Index Kit (Illumina) using a C1000 Touch Thermal Cycler (Bio-rad) programmed with an initial denaturation of 3 min at 95°C, followed by 8 cycles of denaturation at 95°C for 30 sec, annealing at the temperature of 55°C for 30 sec, and extension at 72°C for 5 min. Product concentrations were confirmed using a Qubit (Thermo Fisher Scientific). The libraries were pooled to equal molecular amounts and analyzed on an Illumina Mi-Seq DNA sequencer with a 300 bp paired-end cycle sequencing kit (Illumina). Trimmed reads were mapped to the GreenGenes with default settings using a BaseSpace (16S Metagenomics ver.1.1.0, Illumina).

## Statistical analysis

Comparisons of microbial community structure diversity are analyzed using indicators of α-diversity and β-diversity. The Shannon index was calculated using the 16S Metagenomics application (ver. 1.1.0) within BaseSpace (Illumina). Simpson and Chao1 indicators were calculated using the α-diversity tool in the microbial genomics module of the CLC Genomics Workbench (ver. 23.0.1, Qiagen). All data are described as means ± standard deviation. For the count data of saliva and stool from healthy individuals and UC family members analysed using Illumina BaseSpace, we used ALDEx2 to perform a centred log ratio (CLR) transformation to normalise the count data and compare healthy individuals and UC. For the data after CLR transformation, we performed Welch's t-test (we.ep) and Wilcoxon test (wi.ep) to calculate p-values, and performed multiple test correction (we.eBH, wi.eBH) using the Benjamini-Hochberg method (Fig 2). Statistical analyses were performed by the Log-rank (Mantel-Cox) test (Figs 3 and 4) and Mann-Whitney U test for comparisons between two groups (Figs -4) using GraphPad Prism software v 10.0 (GraphPad Software, San Diego, CA, USA). Statistical significance was set at $p < 0.05$.

Statistical significance was assessed by using the Log-rank (Mantel-Cox) test.

The proportion of *Veillonellaceae* in saliva (b, d) and stool (f, h) samples was shown based on NLR (≥3 vs. <3) and C-reaitive protein (CRP) (≥1 vs. <1). Statistical analyses were performed using a Mann-Whitney U test.

Statistical significance was assessed by using the Log-rank (Mantel-Cox) test.

The proportion of *Prevotellaceae* in saliva (b, d) and stool (f, h) samples was shown based on neutrophil-to-lymphocyte ratio NLR (≥3 vs. <3) and CRP (≥1 vs. <1). Statistical analyses were performed using a Mann-Whitney U test.

## Results

### 1. Disturbances and specific changes in the oral and intestinal and bacterial flora in UC compared to healthy controls

First, we compared the diversity indices of the oral and intestinal bacterial flora using the Simpson index Shannon index, and PCoA. The results were similar between the patients with UC and healthy controls (S1 Fig and S2).

The results of the oral bacterial flora analysis demonstrated significant differences in the relative abundance of the bacterial families *Veillonellaceae*, *Streptococcaceae*, *Prevotellaceae*, and *Neisseriaceae* between patients with UC and healthy controls (Fig 1a). Analysis of the intestinal flora revealed significant differences in the relative abundances of the bacterial families *Veillonellaceae*, *Porphyromonadaceae*, *Prevotellaceae,* and *Ruminococcaceae* between patients with UC and healthy controls (Fig 1b).

Notably, there was a significant increase in *Veillonellaceae* and *Prevotellaceae* were significantly increased in the saliva of patients with UC compared with healthy individuals. (Fig 2). These bacterial floras are particularly elevated in patients with cancer, and it has been suggested that they may influence their response to immunotherapy.

### 2. Relationship between the bacterial flora and immunotherapy efficacy in patients with UC

The effects of specific bacterial flora on OS and PFS were also investigated. It was confirmed that patients with low levels of *Veillonellaceae* in both saliva and stool had longer OS (saliva: P = 0.2094, stool: P = 0.0251) (Fig 3a, b). Similarly, there was a trend towards longer OS in patients with low *Prevotellaceae* levels, but this effect was more pronounced in saliva (saliva: P = 0.0475, stool: P = 0.4712) (Fig 4a, b). In addition, although there was a trend towards prolongation of PFS at low *Veillonellaceae* and *Prevotellaceae* levels in the saliva and stool, the only significant difference was observed for stool *Veillonellaceae* (P = 0.0125) (Fig 3d).

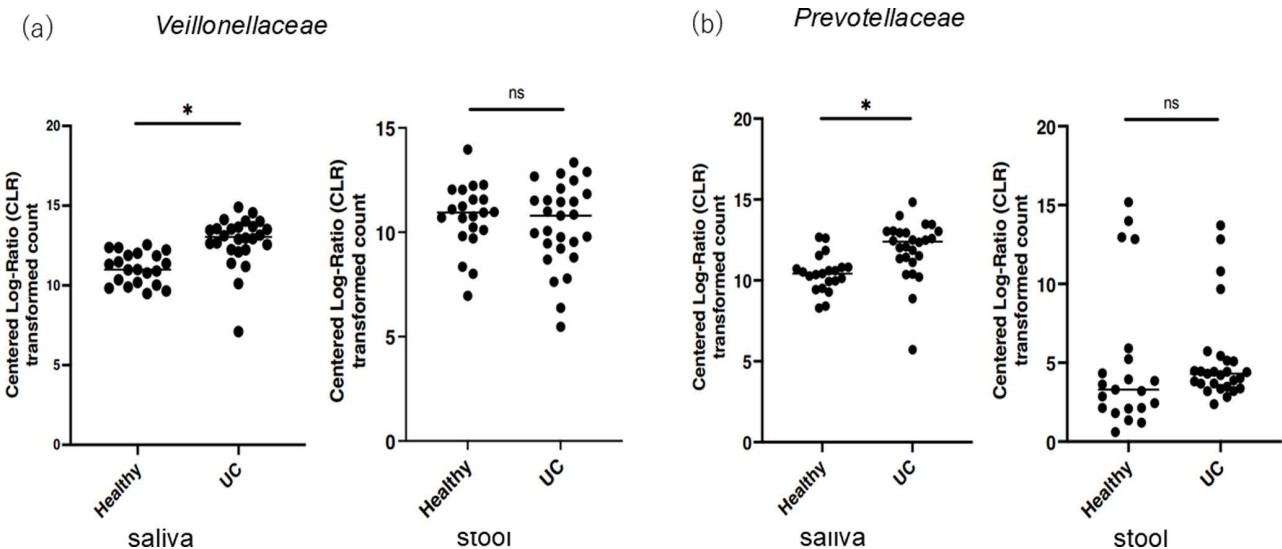

**Fig 2. Comparison of the abundance of *Veillonellaceae* and *Prevotellaceae* in sample groups.** These figures show the abundance of *Veillonella-ceae* (a, b) and *Prevotellaceae* (c, d) in saliva and stool samples from the normal and UC groups. For data after CLR conversion, we calculated p-values using Welch's t-test (we.ep) and Wilcoxon's test (wi.ep), and then performed multiple test correction using the Benjamin-Hochberg method (we.eBH, wi.eBH) to evaluate statistical significance.

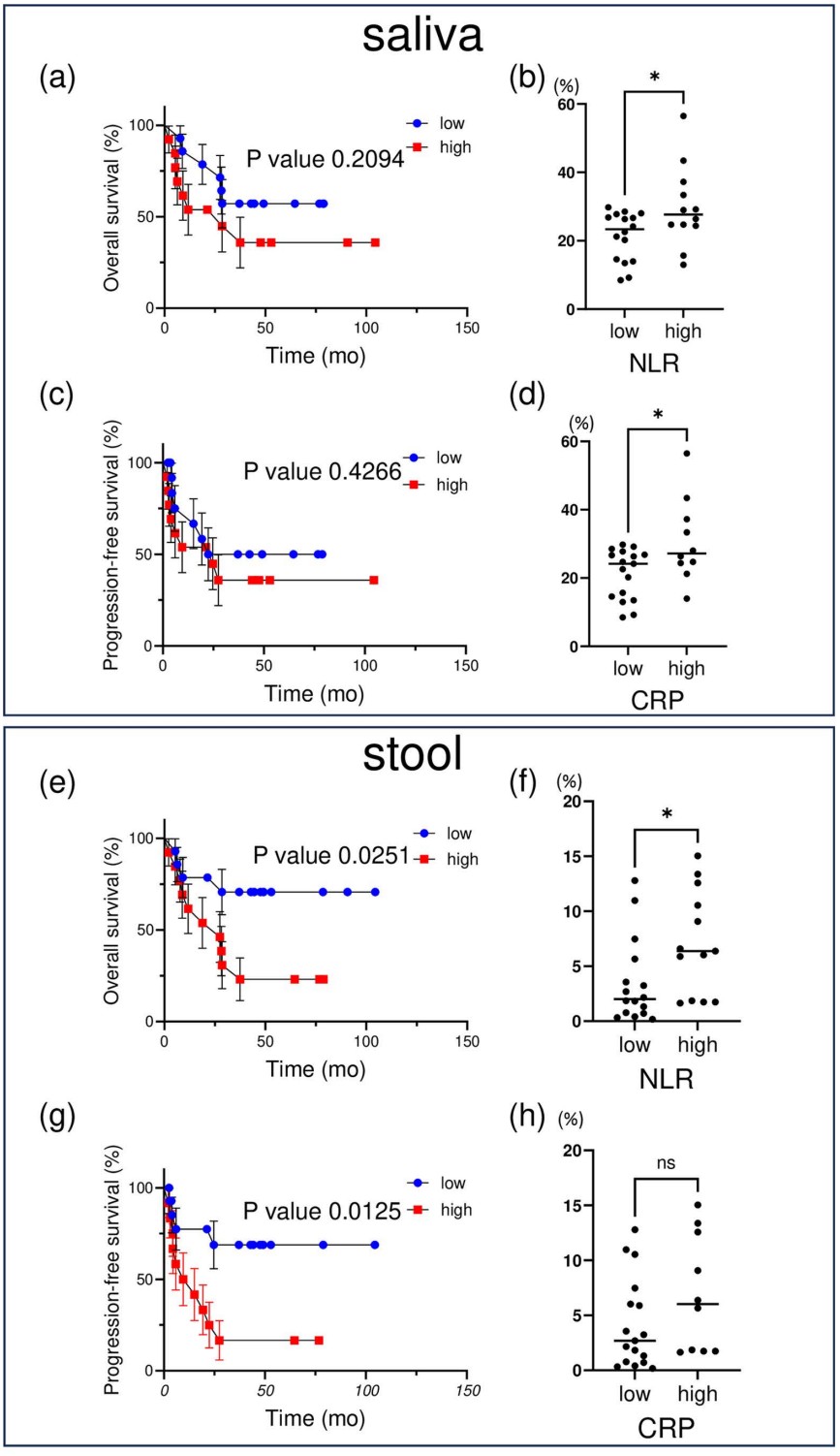

**Fig 3. Relative abundance of *Veillonellaceae* with statistical analysis.** Kaplan-Meier survival curves illustrating the overall survival (OS) and progression free survival (PFS) of UC groups stratified by high and low *Veillonellaceae* abundance in saliva (a, c) and stool (e, g) samples.

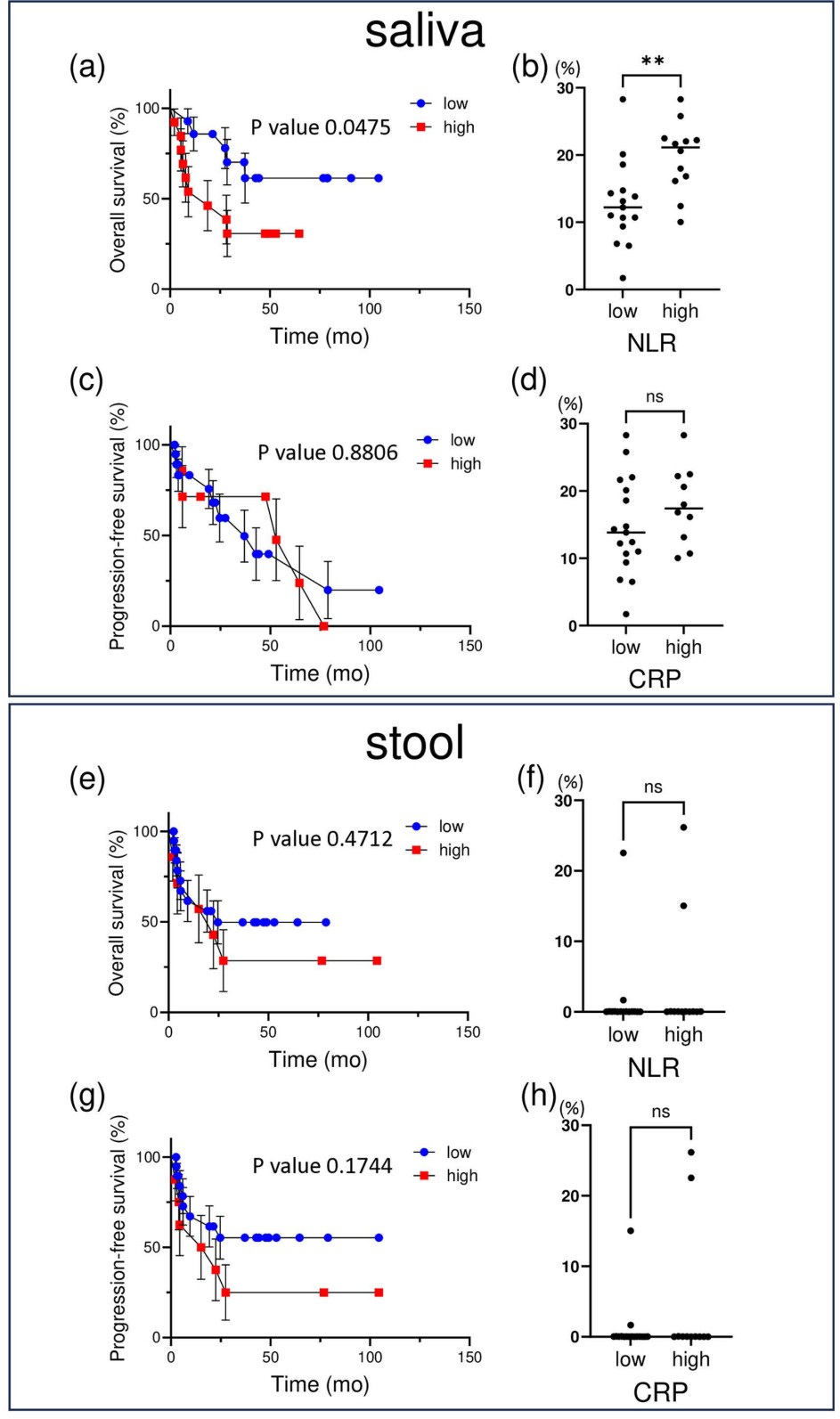

**Fig 4. Relative abundance of *Prevotellaceae* with statistical analysis.** Kaplan-Meier survival curves illustrate the OS and PFS of UC groups stratified by high and low *Prevotellaceae* abundance in saliva (a, c) and stool (b, d) samples.

It was also found that the higher the ratio of both bacterial flora, the higher the NLR. These results suggest that the *Veillonellaceae* to *Prevotellaceae* ratios may be significantly associated with the response to ICIs in patients with UC. They also suggested that specific oral and gut bacteria may improve the response to immunotherapy.

## Discussion

Our findings highlight the critical role of both oral and gut microbiota in modulating the response to ICIs in patients with UC. The observed differences in bacterial composition between patients with cancer and healthy individuals suggest that dysbiosis contributes to carcinogenesis and influences treatment outcomes [1–3]. Specifically, enrichment of *Veillonellaceae* and *Prevotellaceae* in the oral and gut microbiota of patients with UC was strongly associated with survival outcomes. Lower levels of these bacterial families were associated with prolonged OS and PFS, suggesting that elevated levels may negatively affect ICI efficacy.

An important aspect of this study was the relationship between microbiota composition, systemic inflammation, and NLR. A high proportion of *Veillonellaceae* and *Prevotellaceae* was associated with an elevated NLR, which is recognized as an inflammatory marker that reflects systemic inflammation and impaired immune regulation. High NLR is widely considered a poor prognostic factor in cancer because it promotes the secretion of inflammatory cytokines and contributes to an immunosuppressive tumor microenvironment. This may explain the reduced efficacy of ICI observed in patients with UC and the high levels of these bacterial families. In contrast, reduced NLR levels, potentially linked to a healthier microbiome, may enhance immune response, improving tumor control and survival [17,23].

The observed prolonged OS and PFS in patients with low *Veillonellaceae* and *Prevotellaceae* levels support the hypothesis that these bacterial families contribute to systemic inflammation and immunosuppression. Previous studies have suggested that *Veillonellaceae* may promote a proinflammatory environment, exacerbate tumor progression, and reduce the efficacy of immunotherapy. In this context, lower levels of these bacteria may reduce inflammatory signaling and allow for more effective antitumor-immune responses. Similarly, *Prevotellaceae*, particularly in the oral microbiota, have been implicated in systemic inflammation, and their reduction in patients with UC is correlated with improved survival indicators.

Our study is one of the first to examine the combined role of oral and gut microbiota in UC, highlighting the oral microbiota as an equally valuable area of research. Although most microbiome studies have focused on the gut, our findings suggest that oral microbiomes provide important insights into patient prognosis and treatment responses. The association between lower *Prevotellaceae* levels in the oral microbiota and improved OS and PFS highlights the potential of non-invasive oral sampling as a predictive tool for assessing ICI efficacy.

Future research should investigate the mechanistic links between specific bacterial families, NLR, and cancer progression. Understanding how *Veillonellaceae* and *Prevotellaceae* influence systemic inflammation and immune modulation may lead to the development of microbiota-targeted therapies. Such interventions, including probiotics, dietary modulation, or fecal microbiota transplantation, may reduce the NLR and enhance ICI responses, ultimately improving treatment outcomes in patients with UC [3,6].

This study has several limitations. First, the sample size was relatively small, which may have limited the generalizability of the findings. Second, the bacterial composition was not assessed after treatment, preventing analysis of dynamic changes in the microbiota following immunotherapy. Third, although associations with the NLR were identified, the causal relationships between microbiota composition, NLR modulation, and treatment outcomes require further investigation through longitudinal studies.

Our study highlights the significant impact of the oral and gut microbiota on ICI efficacy and cancer outcomes in patients with UC. These results suggest that *Veillonellaceae* and *Prevotellaceae* may serve as biomarkers for predicting survival and treatment responses. Furthermore, the association between these bacterial families, the NLR, and systemic inflammation highlights the potential of microbiota-based therapeutic strategies to optimize immunotherapy outcomes. By

incorporating microbiome profiling into clinical practice, personalized cancer treatment approaches can be developed to improve patient survival and quality of life.

## Supporting information

**S1 Fig. 16S ribosomal gene analysis.** Detailed analysis of microbial composition across additional sample types or conditions. Specific comparisons and statistical comments are provided.
(TIF)

**S2 Fig. Bray-Curtis Distance and PCoA revealing microbial community differences Between healthy and urotherial carcinoma.** Illumina BaceSpace 16S For the count data of the families analysed by metagenomics, we used the vegan R package to calculate the Bray-Cruits distance and then performed intergroup comparisons and PCoA. For intergroup comparisons, we analysed the whole group using PERMANOVA. We also calculated and plotted the distances between Healthy, UC and Healthy-UC. We used the Wilcoxon test to test for significance in each group.
(TIF)

## Acknowledgments

We express our sincere gratitude to the patients and healthy volunteers who participated in this study. We would also like to express our deep appreciation to the staff at the Yamaguchi University Gene Research Center for their technical support and expertise in microbiome analyses. We would like to thank Editage (www.editage.jp) for English language editing.

## Author contributions

**Conceptualization:** Yuichi Matsumoto, Hitaka Yukihiro.

**Data curation:** Yuichi Matsumoto, Hitaka Yukihiro, Keita Kobayashi, Toshio Matsubara, Kenji Watanabe, Yoichi Mizukami, Katsuaki Mishima, Koji Harada.

**Formal analysis:** Yuichi Matsumoto, Hitaka Yukihiro, Yoshiaki Yamamoto, Kenji Watanabe, Yoichi Mizukami.

**Funding acquisition:** Yoshiaki Yamamoto.

**Methodology:** Hitaka Yukihiro, Yoshiaki Yamamoto, Keita Kobayashi.

**Project administration:** Hiroshi Hirata, Yoshiaki Yamamoto, Naohito Isoyama, Koji Shiraishi.

**Resources:** Yuichi Matsumoto, Hitaka Yukihiro, Keita Kobayashi.

**Supervision:** Hitaka Yukihiro, Hiroshi Hirata, Naohito Isoyama, Koji Shiraishi.

**Visualization:** Naohito Isoyama.

**Writing – original draft:** Yuichi Matsumoto, Hitaka Yukihiro, Koji Harada.

**Writing – review & editing:** Koji Shiraishi.

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
