## [Decision Letter · Decision Letter 0]

Dear Dr. Yukihiro,

Thank you for submitting your manuscript to PLOS ONE. After careful consideration, we feel that it has merit but does not fully meet PLOS ONE’s publication criteria as it currently stands. Therefore, we invite you to submit a revised version of the manuscript that addresses the points raised during the review process.

We look forward to receiving your revised manuscript.

Kind regards,

Vinod Kumar Yata, PhD

Academic Editor

PLOS ONE

4. We notice that your supplementary figures are included in the manuscript file. Please remove them and upload them with the file type 'Supporting Information'. Please ensure that each Supporting Information file has a legend listed in the manuscript after the references list.

Additional Editor Comments (if provided):

Reviewers' comments:

Reviewer's Responses to Questions

**Comments to the Author**

1. Is the manuscript technically sound, and do the data support the conclusions?

Reviewer #1: Partly

Reviewer #2: Yes

2. Has the statistical analysis been performed appropriately and rigorously?

Reviewer #1: No

Reviewer #2: Yes

3. Have the authors made all data underlying the findings in their manuscript fully available?

Reviewer #1: No

Reviewer #2: No

4. Is the manuscript presented in an intelligible fashion and written in standard English?

Reviewer #1: Yes

Reviewer #2: Yes

Reviewer #1: Matsumoto et al. describe a brief evaluation of the relationship between microbiome composition and urothelial carcinoma in a unique dataset containing data on both the gut and oral microbiota of patients undergoing treatment. While there are minor suggestions that could be made (the abstract does not adequately justify why the oral microbiome would be relevant to the urinary tract, for example, though it is covered in the introduction), I believe there are several major issues that are much more important to address.

Line 36: The abstract mentions using QIIME to generate operational taxonomic units, but OTUs are not mentioned anywhere in the manuscript. It appears all other analyses use web portal-based tools from Illumina; this discrepancy should be clarified.

Line 158: The methods used for assessing compositional differences between groups are insufficiently described and supported here. My primary objection is in using the Student's t-test to evaluate differences in relative abundance—it's unclear whether relative abundance was tested using read counts, percentages, or some kind of transformed count, but figure 2 suggests the test was done on percentages. I am very skeptical that this is an appropriate method to use, as percentages within a sample are not independent: if one taxon grows significantly, the relative abundance of at least one other taxon, and likely many others, will fall. There are dozens of relevant papers debating the appropriate methods for differential abundance testing (e.g. https://doi.org/10.3389/fmicb.2017.02224); if the authors prefer to use the Student's t-test on relative abundances, I strongly believe it necessary that they provide references for why this is appropriate.

160: This test is described in multiple places as the "non-parametric Student's t test"; the only Student's t-test I am aware of is certainly parametric.

163: The manuscript states a significance threshold of 0.05, but does not specify how many tests were performed nor whether multiple test correction was applied. This is critical information: Figure 2 describes four tests covering two taxa, but presumably there were far more tests performed. If all families were tested for between-group differences, I would guess that even a generous approach to multiple test correction would alter most of the described p-values to be far above the set significance threshold. It is impossible for me to estimate any more accurately because this information was not available, but I strongly believe these results are invalid without a consideration of multiple testing and a description of how it was performed. This should be added.

I leave it to the editor to evaluate the authors' adherence to PLOS editorial policies, but I should note that the manuscript does not describe any deposition of sequencing data, and if any other data was submitted as supplementary files, I was unable to see them. As presented, the analysis described is impossible to replicate. I also am surprised to find that dozens of subjects were recruited, then samples were collected, prepped, sequenced and analyzed, without any disclosed funding.

I hesitate to submit a review that sounds this negative: The authors have collected an interesting dataset and describe compelling reasons for evaluating the microbiome in the context of this disease. However, it is vital that the methodology be updated to be more in line with accepted standards, or at least methods supported by literature, to avoid misleading patients and healthcare practitioners who may look to papers like this for clues to treatment.

Reviewer #2: Line 81-83: Please define your specific aim more precisely.

Lines 86-88: The introduction's mention of Veillonellaceae and Prevotellaceae may suggest predetermined conclusions rather than objective investigation. Please reconsider this framing.

Lines 112 & 114: Please correct the sentence structure to eliminate repetition.

Line 350: There is an inconsistency with Line 170. Figure 1 should present alpha diversity (Simpson and Shannon indices). Please correct this discrepancy.

The methodology section lacks information on raw data processing. Please specify:

1) The number of reads per sample after processing

2) Whether samples were sequenced with adequate depth

3) Analysis of overall community differences (beta diversity)

4) Visualization of beta diversity through PCoA plots

**Do you want your identity to be public for this peer review?** For information about this choice, including consent withdrawal, please see our Privacy Policy

Reviewer #1: No

Reviewer #2: No

---

## [Author Response · Author response to Decision Letter 1]

11 Apr 2025

**Point-by-point responses to the Reviewers’ comments**

Reviewers’ comments:

Reviewer #1:

Line 36: The abstract mentions using QIIME to generate operational taxonomic units, but OTUs are not mentioned anywhere in the manuscript. It appears all other analyses use web portal-based tools from Illumina; this discrepancy should be clarified.

Response:

Thank you for your comment. We did not use QIIME but Illumina’s web portal-based tool. We have corrected the sentence accordingly (lines 38–39 of the revised manuscript).

Line 158: The methods used for assessing compositional differences between groups are insufficiently described and supported here. My primary objection is in using the Student's t-test to evaluate differences in relative abundance—it's unclear whether relative abundance was tested using read counts, percentages, or some kind of transformed count, but figure 2 suggests the test was done on percentages. I am very skeptical that this is an appropriate method to use, as percentages within a sample are not independent: if one taxon grows significantly, the relative abundance of at least one other taxon, and likely many others, will fall. There are dozens of relevant papers debating the appropriate methods for differential abundance testing (e.g. https://doi.org/10.3389/fmicb.2017.02224); if the authors prefer to use the Student's t-test on relative abundances, I strongly believe it necessary that they provide references for why this is appropriate.

Response:

Thank you for your comments on the certification. We used the centred log-ratio (CLR) transformation to normalise the count data of saliva and stool from healthy individuals and UC family members. We have clarified this method in lines 177–181 of the revised manuscript.

160: This test is described in multiple places as the "non-parametric Student's t test"; the only Student's t-test I am aware of is certainly parametric.

Response:

Thank you for your comment. We have corrected the name of the test as “Mann-Whitney U test” (line 185 of the revised manuscript).

163: The manuscript states a significance threshold of 0.05, but does not specify how many tests were performed nor whether multiple test correction was applied. This is critical information: Figure 2 describes four tests covering two taxa, but presumably there were far more tests performed. If all families were tested for between-group differences, I would guess that even a generous approach to multiple test correction would alter most of the described p-values to be far above the set significance threshold. It is impossible for me to estimate any more accurately because this information was not available, but I strongly believe these results are invalid without a consideration of multiple testing and a description of how it was performed. This should be added.

Response:

Thank you for your comments on the certification. We have clarified that we performed multiple test correction using the Benjamini-Hochberg method in lines 181–184 of the revised manuscript.

I leave it to the editor to evaluate the authors' adherence to PLOS editorial policies, but I should note that the manuscript does not describe any deposition of sequencing data, and if any other data was submitted as supplementary files, I was unable to see them. As presented, the analysis described is impossible to replicate. I also am surprised to find that dozens of subjects were recruited, then samples were collected, prepped, sequenced and analyzed, without any disclosed funding.

Response:

Thank you for your comments on the certification. As suggested, we have described the deposition of sequencing in lines 187–188 of the revised manuscript. We have added our funding sources.

Reviewer #2:

Lines 81-83: Please define your specific aim more precisely.

Response:

Thank you for your comment. We have defined our specific aim more precisely in lines 85–89 of the revised manuscript.

Lines 86-88: The introduction's mention of Veillonellaceae and Prevotellaceae may suggest predetermined conclusions rather than objective investigation. Please reconsider this framing.

Response:

Thank you for pointing out the concern about the mention of the Veillonellaceae and Prevotellidae families. We have deleted this text as it lacks objectivity.

Lines 112 & 114: Please correct the sentence structure to eliminate repetition.

Response:

As pointed out, we corrected the structure of the sentence and eliminated the duplication.

Line 350: There is an inconsistency with Line 170. Figure 1 should present alpha diversity (Simpson and Shannon indices). Please correct this discrepancy.

Response:

Thank you for pointing out the discrepancy. We have corrected the figure number as Figure S1 (line 222 of the revised manuscript).

The methodology section lacks information on raw data processing. Please specify:

1) The number of reads per sample after processing

2) Whether samples were sequenced with adequate depth

3) Analysis of overall community differences (beta diversity)

4) Visualization of beta diversity through PCoA plots.

Response:

1) and 2) We have uploaded Excel files (saliva_reads_family.xlsx and stool_reads_family.xlsx) that specified the number of reads per sample and family after processing. These files show that samples were sequenced with adequate depth.

3) and 4) We have analysed the overall community differences and visualized the beta diversity through PCoA plots in Figure S2.

---

## [Editor Report · Decision Letter 1]

Exploration of predictive factors based on oral and intestinal bacterial flora for treating patients with urothelial carcinoma

PONE-D-25-02411R1

Dear Dr. Yukihiro,

We’re pleased to inform you that your manuscript has been judged scientifically suitable for publication and will be formally accepted for publication once it meets all outstanding technical requirements.

Kind regards,

Vinod Kumar Yata, PhD

Academic Editor

PLOS ONE

Additional Editor Comments (optional):

The authors have carefully and comprehensively addressed the reviewers' concerns raised in the initial round of peer review. Specifically:

The statistical methodology has been significantly improved with clarification of the use of centered log-ratio transformation and appropriate non-parametric tests.

The manuscript now includes detailed information on multiple testing corrections using the Benjamini-Hochberg method.

Corrections have been made to statistical terminology (e.g., replacing the inaccurate use of “non-parametric Student’s t-test” with the appropriate “Mann-Whitney U test”).

Details regarding sequencing data availability have been provided, and deposition to GEO (GSE293354) enhances the transparency and reproducibility of the study.

Additional clarifications and improvements to the introduction, aim, methodology, and discussion sections have strengthened the scientific rigor and readability of the manuscript.

The study presents novel and meaningful insights into the association between microbiota and treatment outcomes in urothelial carcinoma, and it aligns well with the scope and readership of PLOS ONE.

On this basis, I am pleased to recommend acceptance of the manuscript in its current form.
---

## [Editor Report · Acceptance letter]

PONE-D-25-02411R1

PLOS ONE

Dear Dr. Yukihiro,

I'm pleased to inform you that your manuscript has been deemed suitable for publication in PLOS ONE. Congratulations! Your manuscript is now being handed over to our production team.

Kind regards,

on behalf of

Dr. Vinod Kumar Yata

Academic Editor

PLOS ONE